# Intermittent fasting induces rapid hepatocyte proliferation to restore the hepatostat in the mouse liver

Abby Sarkar[1]*, Yinhua Jin[1], Brian C DeFelice[2], Catriona Y Logan[1], Yan Yang[3], Teni Anbarchian[1], Peng Wu[1,4], Maurizio Morri[2], Norma F Neff[2], Huy Nguyen[5], Eric Rulifson[1], Matthew Fish[1], Avi Gurion Kaye[1], Azalia M Martínez Jaimes[1], Roel Nusse[1]*

[1]Howard Hughes Medical Institute, Department of Developmental Biology, Institute for Stem Cell Biology and Regenerative Medicine, Stanford University School of Medicine, Stanford, United States; [2]Chan-Zuckerberg Biohub, San Francisco, United States; [3]Stanford Center for Genomics & Personalized Medicine, Stanford University School of Medicine, Stanford, United States; [4]Department of Pediatrics, Stanford University School of Medicine, Stanford, United States; [5]Department of Neurology and Neurological Sciences, Stanford University School of Medicine, Stanford, United States

*For correspondence:
abby.sarkar@gmail.com (AS);
rnusse@stanford.edu (RN)

**Abstract** Nutrient availability fluctuates in most natural populations, forcing organisms to undergo periods of fasting and re-feeding. It is unknown how dietary changes influence liver homeostasis. Here, we show that a switch from ad libitum feeding to intermittent fasting (IF) promotes rapid hepatocyte proliferation. Mechanistically, IF-induced hepatocyte proliferation is driven by the combined action of systemic FGF15 and localized WNT signaling. Hepatocyte proliferation during periods of fasting and re-feeding re-establishes a constant liver-to-body mass ratio, thus maintaining the hepatostat. This study provides the first example of dietary influence on adult hepatocyte proliferation and challenges the widely held view that liver tissue is mostly quiescent unless chemically or mechanically injured.

## Editor's evaluation

This work reports that intermittent fasting alters the homeostatic regenerative programme with fundamental implications for the use of murine models to study liver regeneration and cancer and highlights through a series of solid mechanistic studies the role of FGF/Wnt signalling interactions in modulating fasted-associated regeneration. It opens up further questions as to why this occurs, how this is beneficial to adapting to a fasting state and how we should design and interpret preclinical animal studies.

## Introduction

Periods of fasting and re-feeding induce profound tissue remodeling and regeneration in several tissues including the intestine (*O'Brien et al., 2011*; *Yilmaz et al., 2012*), the muscle (*Cerletti et al., 2012*), and blood (*Brandhorst et al., 2015*; *Chen et al., 2003*; *Ertl et al., 2008*). These tissue changes are thought to be mediated through diet-induced growth factor signaling, including both local (paracrine) and systemic (endocrine) signals that influence cell biology and function (*Mihaylova et al., 2014*). The impact of fasting and re-feeding on liver tissue homeostasis is unknown.

In contrast to other organs, the liver maintains a constant ratio with body weight to preserve homeostasis—this is termed the hepatostat (*Michalopoulos, 2021*). For example, when injured, the liver restores this ratio through hepatocyte renewal, resulting in the liver's ability to maintain its many metabolic functions that are executed by hepatocytes. Organized into hexagonal lobular units, hepatocytes are stacked in between a central vein and a portal triad, which consists of a portal vein, hepatic artery, and bile duct. The directional flow of oxygenated blood from the portal to central axis creates a gradient of cytokines, nutrients, and growth factors throughout the liver lobule that influences hepatocyte transcriptome and function (*Halpern et al., 2017*). Thus, pericentral hepatocytes, present near the central vein, receive different growth factor signals compared to the midlobular and periportal hepatocytes that occupy the rest of the liver lobule. Hepatocyte turnover along the liver lobule has been well characterized during ad libitum (AL) feeding (*Chen et al., 2020*; *He et al., 2021*; *Lin et al., 2018*; *Wang et al., 2015*; *Wei et al., 2021*), when animals are given constant access to food. In these studies, hepatocytes have a detectable rate of turnover and division (*He et al., 2021*; *Wei et al., 2021*), but the proliferation rates compared to other tissues are low (*Michalopoulos, 2021*). In contrast, no study so far has looked at hepatocyte turnover during periods of fasting and re-feeding, arguably a dietary state that more closely mimics nutrient availability and intake in natural populations, where periods of food availability fluctuate.

## Results

### Rapid proliferation of pericentral hepatocytes occurs during intermittent fasting

To determine the impact of intermittent fasting (IF) on hepatocyte turnover, we compared the spatial expression of the proliferation marker Ki67, in 1- and 3-week IF-treated livers compared to AL-treated livers (*Figure 1A–C*). We assessed the presence of Ki67+ hepatocytes throughout the liver lobule using a pericentral hepatocyte marker (glutamine synthetase) and a periportal hepatocyte marker (E-cadherin) (*Figure 1A*). Total Ki67+ hepatocytes increased by 2-fold at 1 week of IF treatment compared to AL treatment (*Figure 1B*). This increase was no longer observed at 3 weeks of IF treatment, suggesting that the increase in proliferation was short term (*Figure 1B*). The number of pericentral Ki67+ hepatocytes increased by approximately 120% after 1 week of IF treatment compared to AL treatment and 3 weeks of IF treatment (*Figure 1C*). At 3 weeks of IF treatment, the number of Ki67+ hepatocytes returned to AL levels, and were predominately midlobular (*Figure 1C*), as has been previously described (*He et al., 2021*; *Wei et al., 2021*).

Next, we corroborated the rapid and positional shift in IF-induced hepatocyte proliferation by employing both random (*Figure 1—figure supplement 1*) and pericentral-specific cell lineage tracing systems (*Figure 1*, *Figure 1—figure supplement 1*) to capture representative hepatocytes and trace their clonal expansion in the liver under an AL or IF feeding regimen. First, to study hepatocyte proliferation throughout the liver lobule, we employed an inducible, *Rosa26-CreERT2* (*Ventura et al., 2007*) allele to permanently and randomly label cells with one of the four fluorophores in the *Rosa26-Confetti* allele (*Snippert et al., 2010*). Following 2 weeks of tamoxifen clearance from the liver (time zero [T0], *Figure 1—figure supplement 1A*), greater than 95% of labeled cells coexpressed one of the fluorophores and the hepatocyte-specific transcription factor HNF4A (*Figure 1—figure supplement 1B*), demonstrating labeling of mostly single hepatocyte clones distributed throughout the liver lobule (*Figure 1—figure supplement 1C*). Three weeks after IF feeding, a distinct increase in pericentral clone size was observed compared to AL-fed animals (*Figure 1—figure supplement 1D*). Second, to study pericentral hepatocyte proliferation kinetics during AL feeding and IF, we utilized an inducible system to mark and trace pericentral hepatocytes. In Axin2-rtTA; TetO-H2B-GFP transgenic mice (*Tumbar et al., 2004*; *Yu et al., 2007*), a modified promoter of the WNT transcriptional target gene, *Axin2*, is used to control expression of a stable histone 2B-GFP fusion protein with doxycycline (dox) administration, thus marking WNT-responsive, pericentral hepatocytes and their progeny (8). Importantly, in these mice the ectopic Axin2-rtTA expression cassette leaves the endogenous *Axin2* locus unchanged. Axin2rtTA; TetO-H2BGFP animals were given dox for 7 days, then cleared of dox for 3 days, and analyzed after dox clearance (T0) or after an additional 6 days of the AL or IF feeding regimen (*Figure 1D*). We observed a 74% increase in GFP-labeled hepatocyte nuclei in IF-treated animals compared T0 animals, thus confirming expansion of pericentral hepatocytes (*Figure 1E, F*).

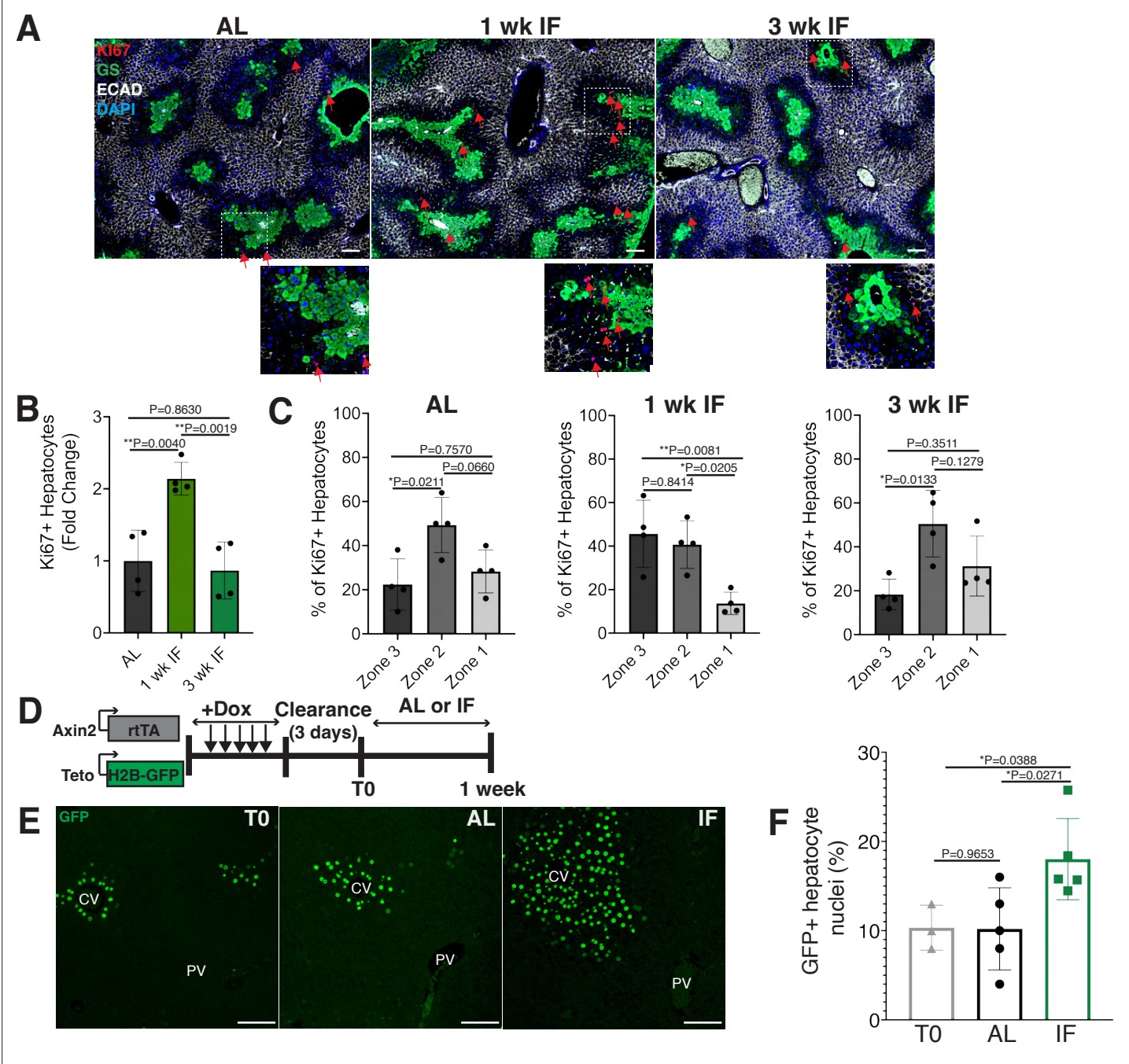

**Figure 1.** Intermittent fasting (IF) induces rapid hepatocyte proliferation. (**A**) Ki67 immunofluorescence for the detection of proliferating cells in ad libitum (AL), 1- and 3-week IF-treated livers. IF-treated livers were analyzed 30 min after re-feeding cycle. (**B, C**) Quantification of spatial distribution and percentage of Ki67+ hepatocytes in AL- and IF-treated livers. One-way analysis of variance (ANOVA), $N$ = 4.(**D**) Dox inducible Axin2-rtTA; Teto-H2BGFP system to label Axin2+ pericentral hepatocytes and trace cell proliferation. Mice were pulsed with dox for 7 days, cleared of dox for 3 days and AL-fed or intermittently fasted for 6 days. (**E**) GFP immunofluorescent images showing increased hepatocyte expansion in AL and IF compared to T0. (**F**) Percentage of GFP+ hepatocyte nuclei in AL, IF livers from A. One-way ANOVA, $N$ = 3 (T0), 5 (AL), 5 (IF). **p < 0.01; *p < 0.05. Error bars indicate standard deviation. Scale bar, 100 µm. wk, weeks.

The online version of this article includes the following figure supplement(s) for figure 1:

**Figure supplement 1.** Hepatocyte proliferation kinetics in ad libitum (AL) fed and intermittent fasted animals.

**Figure supplement 2.** Single-cell RNA-seq comparing hepatocytes in ad libitum (AL) fed and intermittent fasted livers.

No significant change was observed between AL and T0 animals. Additionally, we quantified pericentral hepatocyte proliferation kinetics between 1 week, 3 weeks, and 3 months of IF and AL treatment using Axin2-rtTA; Teto-Cre; Rosa26-mTmG mice (*Figure 1—figure supplement 1E, F*). Notably, the majority of labeled hepatocyte expansion occurred within the first week of IF treatment. Together these data demonstrated the rapid and transient proliferation of pericentral hepatocytes during IF feeding.

As an independent means to characterize hepatocytes after IF treatment, we performed single-cell RNA sequencing on livers from 1-week IF- and AL-treated animals. Livers were collected during a matched, neutral feeding and circadian state. Sequenced hepatocytes were classified into pericentral, midlobular, and periportal hepatocytes (*Figure 1—figure supplement 2A, B*), using well-characterized marker genes (*Halpern et al., 2017*). The proportion of hepatocytes enriched for pericentral transcripts in IF-treated animals was 19.52 ± 4.9%, more than twice as much as the proportion observed in AL-treated animals, 8.97 ± 1.2%. The increased proportion of pericentral hepatocytes in IF-treated animals suggests that IF increases the number of pericentral hepatocytes rather than midlobular or periportal hepatocytes. Furthermore, within pericentral hepatocytes, gene expression analyses revealed a distinct increase in de novo lipogenesis genes (*Fasn*, *Scd1*, and *Acyl*) in IF compared to AL livers (*Figure 1—figure supplement 2C*), highlighting a cellular mechanism for the induction of de novo lipogenesis in the liver, a phenomenon previously observed during IF (*Hatchwell et al., 2020*).

## Nutrient-responsive endocrine FGF15-β-KLOTHO signaling induces hepatocyte proliferation during IF

Endocrine fibroblast growth factor(FGF) signaling is critical in mediating an organism's physiological response to fasting and re-feeding (*Potthoff et al., 2012*). Upon re-feeding, FGF15, produced by intestinal enterocytes and perhaps other tissues too, travels through the bloodstream and binds to its co-receptor, β-KLOTHO (KLB), on hepatocytes (*Inagaki et al., 2005*). Endocrine FGF signaling has also been shown to play important roles in regulating hepatocyte metabolism (*Inagaki et al., 2005*; *Potthoff et al., 2011*) and regeneration (*Uriarte et al., 2013*). To investigate this further, we conducted a kinetic and a histology screen to identify when and where FGF15 acts on hepatocytes during IF. In IF-treated animals, Fgf15 expression was rapidly induced upon re-feeding, reaching maximal levels in the intestine within 30 min upon re-feeding (*Figure 2A*), compared to AL animals which did not detectably express Fgf15 (*Figure 2B*). Thirty minutes after re-feeding, downstream pathway components KLB, PHOSPHO-TYROSINE, and PHOSPHO-C-JUN were elevated in IF animals, and all of these three components were preferentially concentrated in the pericentral region (*Figure 2B*).

Given that endocrine FGF15 is an early regulator of the hepatocyte response to food intake, we asked whether loss of endocrine FGF signaling in hepatocytes would prevent IF-induced proliferation. To test this, we genetically depleted hepatocytes of the endocrine FGF receptor (*Klb*) and traced expansion of Axin2+ GFP-labeled cells during IF treatment (*Figure 2C*). Loss of *Klb* led to a 53% reduction in GFP-labeled pericentral hepatocyte nuclei compared to control animals after 1 week of an IF feeding regimen (*Figure 2D*). Importantly, loss of *Klb* did not impact WNT target gene expression (*Figure 2E*), a critical regulator of hepatocyte zonation and function in the liver (*Perugorria et al., 2019*; *Wang et al., 2015*). Furthermore, loss of *Klb* did not significantly change GFP-labeled nuclei ploidy (*Figure 2F*). In summary, these data emphasize the functional requirement for endocrine FGF15-β-KLOTHO signaling to promote IF-induced pericentral hepatocyte proliferation.

## WNT signaling and the WNT target gene *Tbx3* promote enhanced pericentral hepatocyte proliferation during IF

Next, we sought to understand why pericentral hepatocytes preferentially divided in response to IF, as FGF15 is a hormone and would be theoretically accessible to all hepatocytes. One hypothesis is that IF-induced hepatocyte proliferation additionally requires a second signal, which is concentrated near pericentral hepatocytes. Pericentral hepatocytes receive paracrine WNT signals from endothelial cells of the central vein, which is required to establish and maintain pericentral hepatocyte zonation and function in the liver (*Perugorria et al., 2019*; *Wang et al., 2015*). We tested whether ectopic and constitutive activation of the WNT pathway in midlobular and periportal hepatocytes, which typically do not receive WNT signaling, would promote hepatocyte proliferation under an IF feeding regimen. To ectopically activate WNT signaling, we genetically deleted the WNT repressor *Apc* by injecting

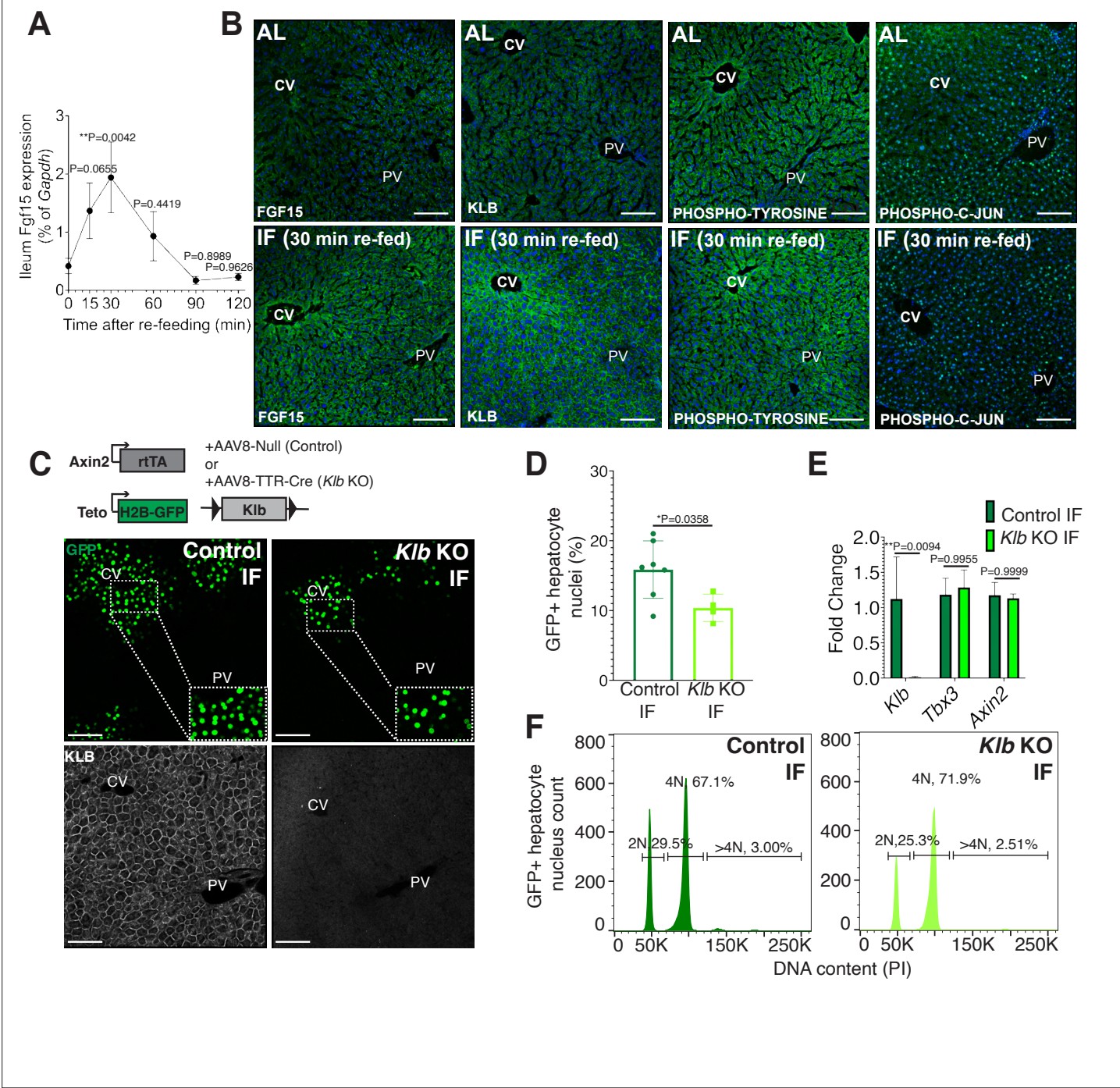

**Figure 2.** Endocrine FGF15-β-KLOTHO (KLB) signaling is required for hepatocyte proliferation during intermittent fasting (IF). (**A**) Quantitative real-time PCR analysis highlighting rapid increase in Fgf15 expression in ileum 30 min after re-feeding in 1-week IF-treated livers. One-way analysis of variance (ANOVA), comparison with time 0, N = 3. (**B**) Immunofluorescence for endocrine FGF pathway components highlighting pathway activation in 1-week IF-treated livers 30 min after re-feeding (ZT12). (**C**) Schematic of method to deplete hepatocytes of *Klb*. Axin2-rtTA; Teto-H2BGFP; *Klb flox/flox* mice were injected with AAV8-TTR-Cre (*Klb* KO). GFP and KLOTHO immunofluorescent images showing decrease in hepatocyte expansion and loss of KLOTHO in *Klb* KO compared to control livers. (**D**) Percentage of GFP+ hepatocyte nuclei in *Klb* KO and control livers. (**E**) Quantitative real-time PCR analysis confirming loss of *Klb* but not WNT target genes, *Tbx3* and *Axin2*, in *Klb* KO livers. Two-way ANOVA, N = 3. (**F**) Ploidy distribution of GFP+ hepatocyte nuclei incontrol IF and *Klb* KO IF livers. Unpaired *t*-test. N = 7 (control), 4 (*Klb* KO). **p < 0.01, *p < 0.05. Error bars indicate standard deviation. Scale bar, 100 μm.

mice ubiquitously expressing *Cas9* with an AAV8 carrying a guide RNA (sgRNA) directed against *Apc* (*Figure 3A*). Midlobular and periportal hepatocytes that had undergone CRISPR-Cas9 gene editing of *Apc* were identified by the expression of WNT target gene glutamine synthetase (GS). Remarkably, these WNT activated (GS+) cells remained mostly as single cells in AL treatment, however they clonally expanded under IF treatment (*Figure 3A, B*).

WNT signaling induces expression of the transcriptional repressor *Tbx3* in hepatocytes. Previous studies have demonstrated that *Tbx3* regulates division of hepatocytes during liver development by repressing cell cycle inhibitors (*Jin et al., 2022*; *Suzuki et al., 2008*). To determine if *Tbx3* plays a role in IF-induced hepatocyte proliferation, we genetically depleted *Tbx3* in hepatocytes and traced expansion of Axin2+ GFP-labeled cells after IF treatment (*Figure 3C*). Loss of *Tbx3* led to a 51% reduction in expansion of GFP-labeled cells after IF treatment (*Figure 3D*). Furthermore, loss of *Tbx3* increased nuclear ploidy in GFP-labeled cells, suggesting that pericentral hepatocytes underwent endoreplication rather than division (*Figure 3E, F*). These findings suggest that WNT and WNT-induced transcription factor TBX3 endow pericentral hepatocytes with capacity to divide during IF treatment.

## FGF15 signaling requires WNT/TBX3 to induce pericentral hepatocyte proliferation

Our results suggest that systemic FGF15 and paracrine WNT pathways may work together to push hepatocytes through the cell cycle. To directly test for an interdependent relationship between FGF and WNT signaling on hepatocyte division, we ectopically expressed *Fgf15* in the liver under AL feeding in the presence or absence of *Tbx3* (*Figure 3G, H*). Indeed, AAV-mediated Fgf15 overexpression, led to a 102% increase in Axin2+ GFP-labeled nuclei compared to control animals, (*Figure 3I*). However, Fgf15 overexpression with *Tbx3* loss did not significantly increase hepatocyte division (*Figure 3I*). Interestingly, Fgf15 overexpression increased nuclear area both with and without loss of *Tbx3* (*Figure 3J*), suggesting that FGF signaling initiates S-phase, but requires WNT through TBX3 to complete mitosis. These results highlight the co-requirement of FGF15 and WNT signaling for pericentral hepatocyte proliferation.

## Hepatocyte proliferation or compensatory polyploidization maintains the hepatostat during IF

During partial hepatectomy where two thirds of the liver is removed or during liver transplantation from a smaller organism to a larger one, reduced liver cell mass relative to overall body size disrupts the hepatostat, the liver-to-body weight ratio required to maintain homeostasis (*Michalopoulos and Bhushan, 2021*). Re-establishment of the hepatostat through hepatocyte regeneration is critical to prevent development of liver disease (*Michalopoulos and Bhushan, 2021*). We asked whether the hepatostat was disrupted during IF. For early timeframes of IF (2–6 days), liver-to-body weight ratio significantly decreased during fasting and increased during re-feeding states compared to AL-treated livers (*Figure 4A*). However, after 3 weeks of IF treatment, this ratio stabilized and was not significantly different between fasting, re-feeding or AL states.

Given the re-establishment of the hepatostat after 3 weeks of IF, we then asked if the transient hepatocyte proliferation observed during early timeframes of IF was required to reach homeostasis after 3 weeks of IF. To test this, we compared the liver-to-body weight ratios, hepatocyte cell and nuclear areas between livers depleted of *Klb* or *Tbx3* and AL-fed or intermittently fasted for 3 weeks. In AL-treated animals, loss of *Klb* did not significantly alter the liver-to-body weight ratio, nor hepatocyte cell and nuclear area compared to control livers (*Figure 4—figure supplement 1A–D*). In contrast, in IF-treated animals, loss of *Klb* led to a significant decrease compared to control livers (*Figure 4B–E*). Livers depleted of *Tbx3* were able to maintain liver-to-body weight ratios, hepatocyte cell and nuclear area similar to control livers during AL and IF treatment (*Figure 4B–E*, *Figure 4—figure supplement 1A–D*). Importantly, IF-treated livers depleted of *Tbx3* exhibited hyper polyploidization of pericentral hepatocytes (*Figure 4D*). These findings highlight the importance of hepatocyte proliferation during IF to maintain the hepatostat. Moreover, they demonstrate the ability of hepatocytes to undergo endoreplication and polyploidization, in the absence of division, as a compensatory mechanism to maintain liver size.

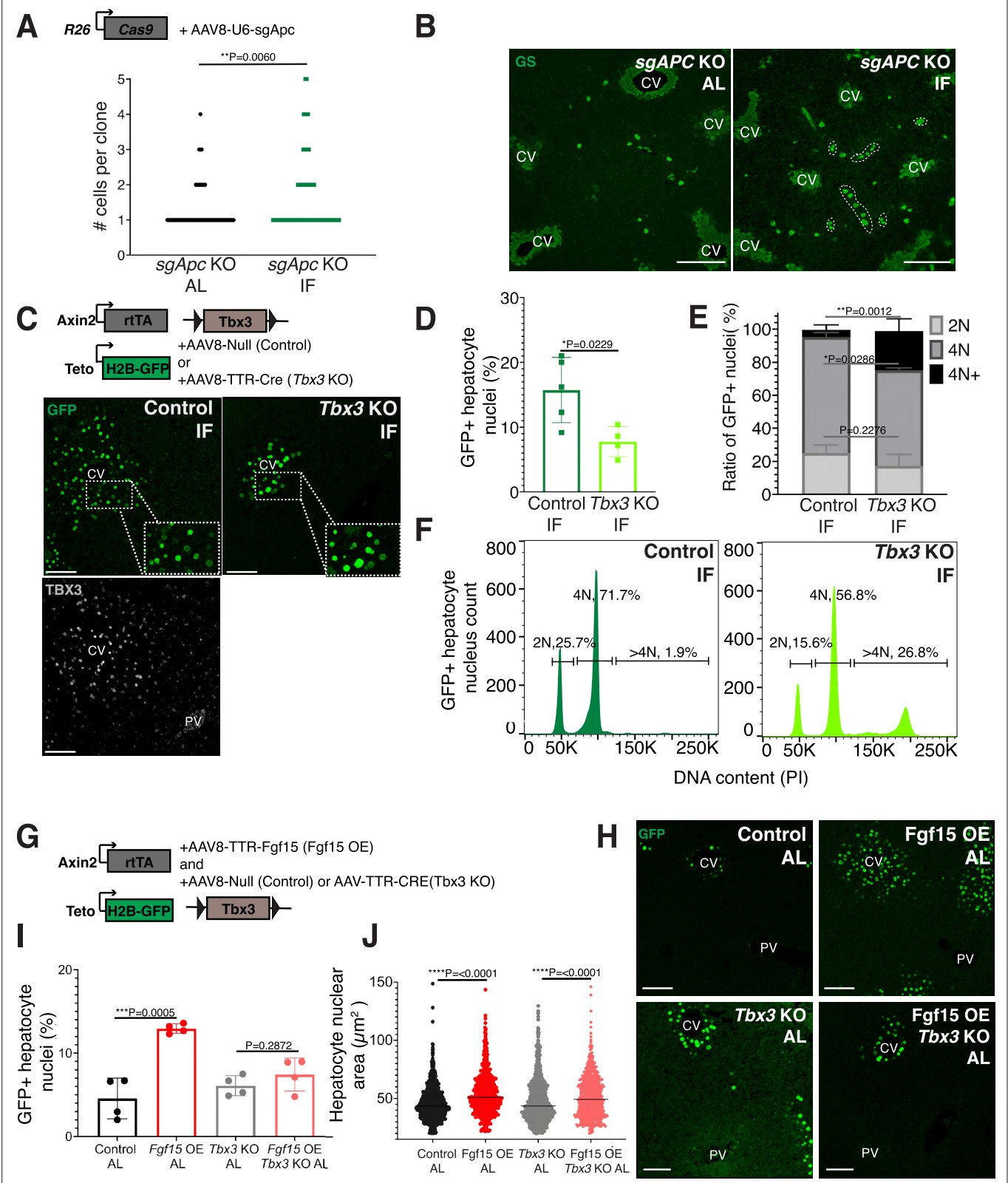

**Figure 3.** Paracrine WNT and WNT target gene *Tbx3* promote hepatocyte proliferation during intermittent fasting (IF). (**A**) Method to constitutively activate WNT signaling in midlobular, periportal cells. AAV8-U6-sgAPC was injected into the tail vein of *Rosa26-Cas9* mice. Animals were IF treated for 1 week before analysis. GS immunofluorescent images for detection of Apc mutant clones in ad libitum (AL) and IF livers. (**B**) The number of Apc mutant hepatocytes per 3D clone expand in IF compared to AL livers. Mann–Whitney test. 130 clones analyzed in AL. 74 clones analyzed in IF. *N* = 3. White

*Figure 3 continued on next page*

*Figure 3 continued*

dashed lines demarcate multicellular non-pericentral GS+ clones. (**C**) Schematic of method to deplete hepatocytes of the WNT target, *Tbx3*. Axin2-rtTA; Teto-H2BGFP; *Tbx3flox/flox* mice were intraperitoneally injected with AAV8-TTR-Cre (*Tbx3* KO). GFP and TBX3 immunofluorescent images to show IF-induced proliferation and *Tbx3* depletion, respectively, in control and *Tbx3* KO livers. (**D**) Percentage of GFP + hepatocyte nuclei decreased in Tbx3 KO IF compared to control IF livers. Unpaired *t*-test, (*N* = 7 control IF), *N* = 4 (*Tbx3* KO IF). (**E, F**) Nuclear ploidy distribution of GFP+ hepatocytes highlighting hyper-polyploidization in *Tbx3* KO IF compared to control IF livers. Two-way analysis of variance (ANOVA), *N* = 3. (**G**) Schematic for Fgf15 overexpression. Axin2-rtTA; Teto-H2BGFP; *Tbx3 flox/flox* mice were injected with AAV-TTR-FGF15 (Fgf15 OE) and AAV8-Null (control) or AAV-TTR-CRE (*Tbx3* KO). (**H**) GFP immunofluorescent images from c AL, Fgf15 OE AL, *Tbx3* KO AL, Fgf15 OE; *Tbx3* KO AL livers. (**I**) Percentage of GFP+ hepatocyte nuclei highlighting lack of hepatocyte proliferation in *Tbx3* KO livers. Unpaired *t*-test, *N* = 4. (**J**) Dot plot highlighting increase in nuclear area with Fgf15 overexpression both with and without *Tbx3*. Unpaired *t*-test. ****$p < 0.0001$, ***$p < 0.001$, **$p < 0.01$, *$p < 0.05$. Error bars indicate standard deviation. Scale bar, 100 μm.

To determine the functional consequence of loss of hepatocyte proliferation during IF, we analyzed fibrosis and cell death in control, *Klb* KO, or *Tbx3* KO livers under IF or AL feeding regimens (**Figure 4—figure supplement 2A–C**). Despite the significant reduction in liver-to-body weight ratio with loss of *Klb* during IF, we did not observe an increase in fibrosis or dying (TUNEL+) hepatocytes with loss of *Klb* or loss of *Tbx3* during IF treatment. However, we did observe a greater than three-fold increase in transaminase AST and ALT—widely used markers of liver injury—in serum from *Klb* KO IF-treated livers compared to control IF and *Tbx3* KO IF-treated livers (**Figure 4F**). We also observed a striking increase in pericentral-localized hepatocyte senescence in *Klb* KO IF-treated livers compared to control IF and *Tbx3* KO IF-treated livers (**Figure 4G**). Furthermore, hepatocyte senescence and serum transaminase levels were elevated by two-fold in *Klb* KO IF-treated livers compared to *Klb* KO AL-treated livers (**Figure 4F, G** and **Figure 4—figure supplement 1E, F**). These two findings demonstrate that when compensatory hepatocyte proliferation is blocked during IF, liver pathology ensues.

To further support these findings, we performed zonation studies and metabolomics to identify additional functional consequences of compromised hepatocyte proliferation during IF. These analyses highlighted marked changes in zonation (**Figure 4H**) as well as liver metabolites (**Figure 4I**) between control IF and *Klb* KO IF-treated livers. Curiously, differences in metabolites between IF and AL samples were lost when *Klb* was lost during IF treatment (**Figure 4J**, **Supplementary file 1**) indicating that the hepatocyte metabolism required to maintain the hepatostat during IF was impaired when hepatocyte proliferation and polyploidization was lost. To further test this, we examined the zonation and expression of genes important in production of bile acids, the most strongly changed metabolites between IF and AL. Importantly, *Cyp7a1*, a gene that is pericentrally expressed in the normal liver (**Halpern et al., 2017**), is dysregulated by threefold with loss of *Klb* during IF treatment (**Figure 4K**). These data combined demonstrate that loss of division of pericentral cells during IF has important functional consequences for the liver including a decrease in the liver-to-body weight ratio, increased pericentral hepatocyte senescence, and irregular liver zonation and metabolism.

## Discussion

The liver is thought to be mostly quiescent except in the presence of injury. Our data overturn this view by demonstrating that the liver is exquisitely tuned to changes in nutrient status and deploys robust homeostatic mechanisms to ensure a constant liver-to-body weight ratio during fasting and re-feeding. In order to proliferate in response to IF, hepatocytes integrate nutrient sensing responses, mediated via FGF15, with knowledge of cellular position, mediated by local, pericentral WNT signals. Pericentral hepatocyte proliferation ensures replacement of lost cellular mass through increases in overall cell numbers. We propose a working model in which paracrine WNT and endocrine FGF pathways work together to push hepatocytes through different phases of the cell cycle. The role of FGF15 in pericentral hepatocyte proliferation observed in our studies suggests that this signal initiates S-phase, whereas WNT through TBX3 permits progression through M-phase. It has been demonstrated in other contexts as well that both FGF and WNT signaling are conjointly required for tissue growth (**McGrew et al., 1997**; **ten Berge et al., 2008**). Whether or not timing of re-feeding and dosage of FGF15, WNT, and TBX3 impact location and degree of hepatocyte proliferation remain important questions for future studies.

It will be of interest to know how the hepatostat is maintained during other nutrient conditions such as ketogenic, calorie restricted, or high-fat diets. Many tissues in the body are known to be

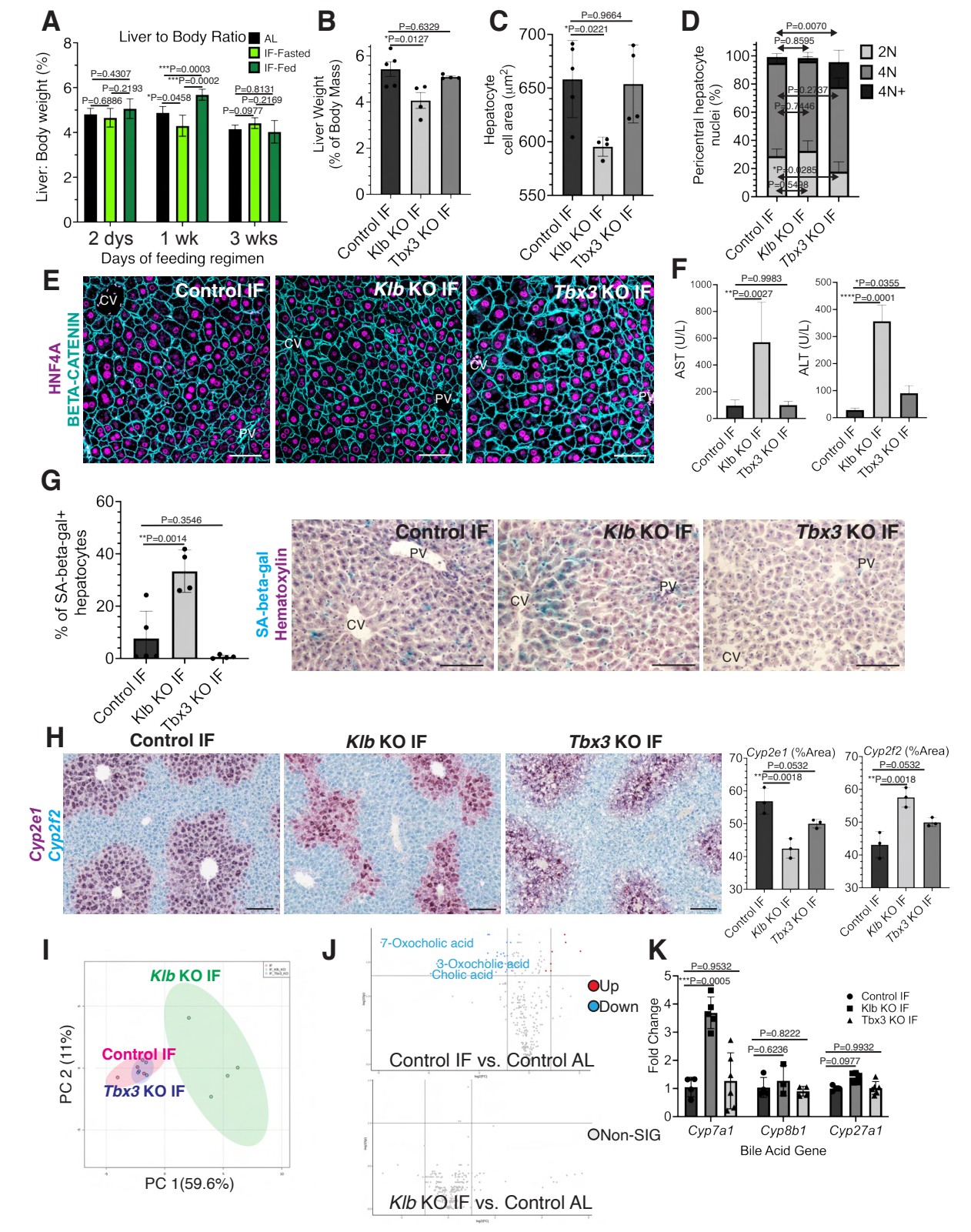

**Figure 4.** Hepatocyte proliferation or compensatory polyploidization maintains the hepatostat during intermittent fasting (IF). (**A**) Liver-to-body weight ratio in wild-type livers during 2 days, 1 week, and 3 weeks of IF and ad libitum (AL) feeding. (**B–K**) Liver analyses after 3 weeks of IF treatment in control, *Klb* KO, and *Tbx3* KO livers. (**B**) Liver-to-body weight ratio. (**C**) Hepatocyte nuclear area. (**D**) Nuclear ploidy distribution of pericentral hepatocytes with hyper-polyploidization in *Tbx3* KO IF livers. (**E**) Immunofluorescence images for β-CATENIN and HNF4A highlighting hepatocyte cell and nuclear area

*Figure 4 continued on next page*

*Figure 4 continued*

during IF. (**F**) AST and ALT liver injury marker presence in serum. (**G**) Quantification and representative images of senescence-associated β-galactosidase stains. (**H**) RNAscope images and quantification for pericentral marker Cyp2e1 and periportal marker Cyp2f2. (**I**) Metabolomics PCA plot comparing control IF, *Klb* KO IF, and *Tbx3* KO IF livers. (**J**) Volcano plots comparing metabolites between control AL and control IF livers and control AL and *Klb* KO IF livers. The top 3 most significantly changed bile metabolites are labeled in blue. (**K**) Expression of bile acid pathway enzymes genes in livers. Quantitative PCR for genes for critical enzymes in bile acid pathway. All statistics were performed on N=3-5 animals using one-way analysis of variance (ANOVA). ****p < 0.0001, ***p < 0.001, **p < 0.01, *p < 0.05. Error bars indicate standard deviation. Scale bar, 100 μm.

The online version of this article includes the following figure supplement(s) for figure 4:

**Figure supplement 1.** Short-term loss of *Tbx3* or *Klb* does not disrupt the hepatostat during ad libitum (AL) feeding.

**Figure supplement 2.** Fibrosis and cell death assessment of Control, *Klb* KO, and *Tbx3* KO IF- and ad libitum (AL)-treated livers.

slowly proliferating, but this conclusion has mainly stemmed from AL-fed mouse studies, which may not mimic the IF that wild animals are exposed to during times of fluctuating food abundance. These studies demonstrate that there is perhaps more proliferative capacity than previously appreciated in tissues that are currently thought to exhibit a slow turnover rate. This would have implications for how we understand regulation of both stem and non-stem cell populations in many other tissues and organs.

## Materials and methods
### Mouse strains, husbandry, and experimental methods

All experiments were done on adult, 8- to 12-week-old, male mice, unless otherwise noted. Wild type *C57BL/6* J mice, *Rosa26-CreERT2* (**Ventura et al., 2007**), *Rosa26-Confetti* (**Snippert et al., 2010**), *Axin2-rtTA* (**Yu et al., 2007**), *TetO-H2B-GFP* (**Tumbar et al., 2004**); *Rosa 26-mTmG* (**Muzumdar et al., 2007Muzumdar et al., 2007**) and *Rosa26-Cas9* (**Platt et al., 2014**) strains were obtained from The Jackson Laboratory (JAX, Bar Harbor, ME, see Key Resources Table). *Tbx3 flox* mice were a gift from Dr. Anne Moon (**Frank et al., 2012**). *Klb* flox mice were a gift from Dr. David Mangelsdorf (**Ding et al., 2012**). All mice were housed in the animal facility of Stanford University on a 12 hr light/dark cycle (0700/1900 hr) AL access to water and food (standard chow diet with 18% calories derived from fat, 24% calories from protein, and 58% calories from carbohydrates, Tekland 2918).

For all IF, mice were randomly assigned into AL or IF groups. IF was performed with total food deprivation and AL access to water from approximately 1900 to 1900 hr the following day to implement alternate periods of 24 hr fasting and feeding. Unless specified otherwise in text and figures, all samples were collected for both AL and IF groups at 1200 hr to access samples during a neutral metabolic and circadian rhythm time point.

For random cell lineage tracing studies, *Rosa26-CreERT2/Rosa26-Conf*etti mice received intraperitoneal injections of tamoxifen (TAM; 4 mg/25 g mouse weight, Sigma, St. Louis, MO) dissolved in 10% ethanol/corn oil (Sigma, St. Louis, MO) twice with 48 hr between injections. Two weeks after the last tamoxifen injection, livers where immediately analyzed (T0) or analyzed after an additional 1–3 weeks of AL feeding or IF.

For Axin2+ cell lineage tracing studies, mice received doxycycline hyclate (Dox; 1 mg/ml; Sigma, St. Louis, MO) in drinking water for 5 days. Dox water was then replaced with normal drinking water for 3 days before livers were immediately analyzed (T0) or analyzed after an additional 1 week, 3 weeks, and 3 months of AL feeding or IF.

For all AAV studies, mice were intraperitoneally injected with $1 \times 10^{11}$ genome copies per mouse at 6–8 weeks of age to induce liver-specific depletion of *Klb* or *Tbx3* (AAV8-TTR-Cre, Vector Bio Labs, Malvern, PA), Fgf15 overexpression (AAV8-TTR-FGF15; Addgene deposit 81516), or *Apc* gene editing (AAV8-U6-sgAPC; derived from Clontech 632609). For AAV control studies, an AAV8-Null vector containing no transgene (Vector Bio Labs, Malvern, PA) was used on a combined cohort of *Tbx3 flox/flox* and *Klb flox/flox* mice. For studies that combined AAV injection and Axin2+ cell lineage tracing, mice were first injected with AAV, allowed 3 days to recover and subsequently treated with dox water to induce tracing.

All animal experiments and methods were approved by the Institutional Animal Care and Use Committee at Stanford University. In conducting research using animals, the investigators adhered to the laws of the United States and regulations of the Department of Agriculture.

## Tissue collection, processing, staining, and imaging

For clonal analysis and KLOTHO immunofluorescence, mice were perfused with 4% paraformaldehyde (PFA), livers were isolated and further fixed in 4% PFA for 2 hr at 4°C. PFA-fixed tissues were washed in phosphate-buffered saline (PBS) and sectioned into 50–200 µm sections using a Compresstome vibrating microtome tissue slicer (VF-310-0Z, Precisionary). Vibratome sections from the median lobe were then permeabilized, stained and cleared using a method developed by the Zerial lab (https://www.zeriallab.org/). Antibodies and dilutions described in Key Resources Table. Vibratome sections were imaged using an SP8 White Light Laser Confocal microscope (Lecia, Weltzar, Germany) and a BZ-X800 microscope (Keyence, Osaka, Japan). Confocal image stacks were acquired at ×20 magnification and up to 100 µm with a step size of 1 µm along the Z-axis and processed and analyzed with Imaris software.

For endocrine FGF signaling pathway analyses (*Figure 2B*), livers were flash frozen in OCT, cryosectioned at 10 µm, fixed for 15 min in 4% PFA at room temperature and then stained. Antibodies and dilutions described in Key Resources Table.

For histology and immunohistochemistry, liver was fixed overnight in 10% formalin at room temperature, dehydrated, cleared in HistoClear (Natural Diagnostics), and embedded in paraffin. Sections were cut at 5 µm thickness, de-paraffinized, re-hydrated, and processed for further staining via immunofluorescence or in situ hybridization assays as described below.

For histology, formalin-fixed paraffin-embedded liver sections were sent to the Department of Comparative Medicine's Animal Histology Services for Sirius Red staining.

For immunofluorescence, sections of formalin-fixed paraffin-embedded livers were subjected to antigen retrieval with Tris buffer pH = 8.0 (Vector Labs H-3301, Newark, CA) in a pressure cooker. They were then blocked in 5% normal donkey serum in PBS containing 0.1% Triton X, in combination with the Avidin/Biotin Blocking reagent (Vector Labs SP-2001, Newark, CA). Sections were incubated with primary and secondary antibodies and mounted in Prolong Gold with DAPI medium (Invitrogen, Waltham, MA). Biotinylated goat antibody was applied to section stained with TBX3, before detection with Streptavidin-647. Antibodies and dilutions described in Key Resources Table. Samples were imaged at ×20 magnification using an Sp8 Confocal or a Zeiss Imager Z.2 and processed and analyzed with ImageJ software.

## Senescence-associated β-galactosidase staining and TUNEL assay

For senescence-associated β-galactosidase staining, flash frozen livers were cryo-sectioned at 10 µm and fixed with 0.5% glutaraldehyde in PBS for 15 min, washed with PBS supplemented with 1 mM $MgCl_2$ and stained for 14 hr in PBS containing (1 mM $MgCl_2$; 1 mg/ml X-Gal and 5 mM of each of potassium ferricyanide and potassium ferrocyanide). Assay was performed at pH = 5.5 as previously described (*Krizhanovsky et al., 2008*). Hematoxylin was used as a counterstain.

To detect apoptosis in livers, formaldehyde-fixed paraffin-embedded sections were detected by terminal deoxynucleotidyl transferase-mediated dUTP-biotin nick end labeling (TUNEL) assay (Thermo Fisher Scientific, C10619, Waltham, MA) according to the manufacturer's instructions.

## RNAscope in situ hybridization

In situs were performed using the RNAscope 2.5 HD Duplex Reagent Kit (Advanced Cell Diagnostics, Newark, CA) according to the manufacturer's instructions. Images were taken at ×20 magnification on a Zeiss Imager Z.2 and processed using ImageJ software. Probes used in this study were *Cyp2f2* (target region: 555–1693) and *Cyp2e1* (target region: 458–1530).

## Fibrosis assay

Bright-field images were collected on a Zeiss Imager Z.2. Red stained collagen levels were quantified using ImageJ (imagej.nih.gov).

## Clone size, number, and location

To quantify clone size, threshold of fluorescent channels was lowered so that clear cell and nuclear boundaries could be distinguished. One large-stitched image with an area of 1.8 × 1.8 × 0.1 mm³ mouse was taken from two representative vibratome sections from each mouse at each time point. Only clones completely within the tissue sample were analyzed. We counted the total number of

clones from the six representative images. Because images were of equal area, clone numbers can be compared to each other between time points. Clones containing a cell located within three cell distances from the portal vein or bile duct were classified as periportal; clones containing a cell within three cell distances from the central vein were classified as pericentral; clones not meeting either criterion were classified as midlobular. For clone size quantification in the AAV8-U6-sgAPC model, only GS+ clones that had discrete boundaries, one or more GS-negative hepatocytes from surrounding GS+ pericentral cells, were quantified .

## Hepatocyte nuclei isolation and analysis

For hepatocyte nuclei isolation, liver lobes from mice were homogenized in cold 1% formaldehyde in PBS with a loose pestle and Dounce homogenizer. Samples were then fixed for 10 min at room temperature followed by incubation for 5 min with glycine at a final concentration of 0.125 M. Samples were centrifuged at 300 × $g$ for 10 min, at 4°C. Pellets were washed in PBS and re-suspended with 10 ml cell lysis buffer (10 mM Tris–HCl, 10 mM NaCl, 0.5% IGEPAL) and filtered through 100 µm cell strainers. A second round of homogenization was performed by 15–20 strokes with a tight pestle. Nuclei were pelleted at 2000 × $g$ for 10 min at 4°C and re-suspended in 0.5 ml PBS and 4.5 ml of pre-chilled 70% ethanol and stored at −20°C before downstream GFP content and ploidy analysis by flow cytometry.

Right before flow cytometry, 1 million nuclei were re-suspended in PBS and stained with FxCycle PI/RNase (Thermo Fisher, F10797, Waltham, MA) staining solution for 15–30 min at room temperature. Cells were analyzed on an FACS ARIA II (BD). Data were processed with FACS Diva 8.0 software (BD) and FlowJo v10 (FlowJo). Doublets were excluded by FSC-W × FSC-H and SSC-W × SSC-H analysis. Single-stained channels were used for compensation and fluorophore minus one control was used for gating.

## Real-time PCR measurement

Liver samples were homogenized in TRIzol (Invitrogen, Waltham, MA) with a bead homogenizer (Sigma, St. Louis, MO). Total RNA was purified using the RNeasy Mini Isolation Kit (Qiagen, Hilden, Germany) and reverse transcribed (High Capacity cDNA Reverse Transcription Kit; Life Technologies, Carlsbad, CA) according to the manufacturer's protocol. Quantitative RT-PCR was performed with TaqMan Gene Expression Assays (Applied Biosystem, Waltham, MA) on an StepOnePlus Real-Time PCR System (Applied Biosystems, Waltham, MA). Relative target gene expression levels were calculated using the delta-delta CT method (*Livak and Schmittgen, 2001*). Gene Expression Assays used were *Gapdh* (Mm99999915_g1) as control, *Klb* (Mm00473122_m1), *Axin2* (Mm00443610_m1), *Fgf15*(Mm00433278_m1), and *Tbx3* (Mm01195719_m1) all from Thermo Fisher Scientific (Waltham, MA).

## Single-cell RNA sequencing

Hepatocytes were isolated from livers of 8-week-old C57BL/6J mice that had been intermittent fasted for 1 week or AL-fed using a two-step collagenase perfusion technique as previously described (*Peng et al., 2018*).

Collections were performed at 1000 hr and during the feeding cycle of IF. For each feeding regimen, three livers were collected and processed as three individual samples. For each sample, 2000 hepatocytes were loaded to target ~1000 cells after recovery according to the manufacturer's protocol. Single cell libraries were prepared using the 10× Genomics Chromium Single Cell 3″ Reagents Kit V3. Single-cell libraries were loaded on an Illumina NovaSeq 6000 instrument with NovaSeq S2 v.1.5 Reagent Kits with the following reads: 28 bases Read 1 (cell barcode and unique molecular identifier [UMI]), 8 bases i7 Index 1 (sample index), and 91 bases Read 2 (transcript).

Sample demultiplexing, barcode processing, single-cell counting, and reference genome mapping were performed using the Cell Ranger Software (v3.1.0, mm10 ref genome) according to the manual. All samples were normalized to present the same effective sequencing depth by using Cell Ranger aggr function. The dimensionality reduction by principal components analysis (PCA), the graph-based clustering and UMAP visualization were performed using Seurat (v3.0, R package). Genes that were detected in less than three cells were filtered out, and cells were filtered out with greater than 10% of mitochondrial genes and with fewer than 200 or greater than 50,000 detected genes.

For cell clustering, R software was used to sort cells into either pericentral (PC), midlobular (Mid), and periportal (PP) classes based on the greatest expression of biomarkers *Cyp2e1*, *Cyp1a2*, *Glul* (for PC), *Hamp* and *Igfbp2* (for Mid), and *Cyp2f2* and *Cps1* (for PP).

## Generation of AAV expression vectors

The AAV-TTR-FGF15 virus was produced from the complementary stand AAVS construct, csAAV-TTR-CRE plasmid (kind gift of Holger Willenbring). The *CRE* gene was excised by digestion with SalI (NEB). The *Fgf15* gene (GenBank: BC021328 cloneID 5066286) was amplified for assembly into the SalI cut AAV-TTR backbone with NEB HiFi Builder using the primers: (FWD) 5′ ggagaagcccag ctgGTCGACGCCACCATGGCGAGAAAGTGGAACGG 3′ and (REV) 5′ atcagcgagctctaGTCGACTCAT TTCTGGAAGCTGGGACTCTTCAC 3′. The two fragments were assembled with NEB HiFi builder and cloned in NEB Stable *E. coli*.

The AAV-sgApc virus was produced from the pAAV-Guide-it-Down construct (Clontech Laboratories Inc, 041315) using assembly primers:

> (FWD) 5′CCGGAGGCTGCATGAGAGCACTTG3′ and
> (Rev) 5′AAACCAAGTGCTCTCATGCAGCCT3′.
>
> AAV-sgApc contains a U6 promoter and an sgRNA targeting the sequence 5′AGGCTGCA TGAGAGCACTTG3′ in exon 13 of *Apc*.

## Metabolite extraction

Livers were harvested and immediately flash frozen in LN2 then stored at −80°C. While kept on dry ice a 20-mg sample was removed from each liver specimen, massed using an analytical balance, and placed in a 2-ml round bottom polypropylene tube containing 4–6, 2.3-mm stainless steel beads. 500 µl of −20°C extraction solution (methanol:acetonitrile:water, 2:2:1) containing stable isotope-labeled metabolite standards was added to each sample tube. Ratio of 20 mg to 500 µl was retained when masses were not exactly 20 mg. All samples were homogenized at an amplitude of 20 Hz for 15 min and stored at −20°C for 1 hr to maximize protein precipitation. Samples were then vortexed for 20 s and centrifuged at 4°C for 5 min, speed 14,000 rcf. 120 µl of supernatant was removed from each tube and filtered using 0.2-µm polyvinylidene fluoride filter (Agilent Technologies P/N: 203980-100) and collected via 6000 rcf centrifuge for 4 min. An additional 50 µl was removed from each sample and combined into five pooled samples analyzed at equal intervals throughout the analysis to ensure stable signal. Extracts, pools, and procedural blanks were sealed and stored at 4°C until prompt analysis.

## HILIC–MS/MS metabolite data collection and processing

Untargeted metabolomics analysis was conducted as described previously (*Han et al., 2021*) with some modification. Liver extracts were analyzed via hydrophilic interaction liquid chromatography (HILIC) coupled to a Thermo Q-Exactive HF high resolution mass spectrometer. Each sample was analyzed in both positive and negative ionization modes (ESI+, ESI−) via subsequent injections. Full MS-ddMS2 data were collected, an inclusion list was used to prioritize MS2 selection of metabolites from our in-house 'local' library, when additional scan bandwidth was available MS2 was collected in a data-dependent manner. Mass range was 60–900 *m/z*, resolution was 60 k (MS1) and 15 k (MS2), centroid data were collected, loop count was 4, isolation window was 1.5 Da. Metabolomics data were processed using MS-DIAL v4.60 (*Tsugawa et al., 2020*) and queried against a combination of our in-house MS2 library (*Han et al., 2021*) and MassBank of North America, the largest freely available spectral repository (*Kind et al., 2018*). Annotations were scored using guidelines from the metabolomics standards initiative (*Sansone et al., 2007*). Features were excluded from analysis if peak height was not at least fivefold greater in one or more samples compared to the procedural blank average. Statistical analysis of annotated features was implemented using MetaboAnalyst 5.0 (*Pang et al., 2021*). Data visualization including principal component analysis and volcano plots were generated using log10 transformed peak heights.

## Acknowledgements

We thank KM Loh and N Torok for constructive feedback on the manuscript; the Chan Zuckerberg Biohub Community Access Program for use of 10× Genomics equipment and sequencing; Stanford Neuroscience Gene and Vector Virus Core for virus production; and H Willenbring for sharing the csAAV-TTR-CRE plasmid. This study was supported by the Howard Hughes Medical Institute (HHMI) and the Stinehart Reed Foundation. AS was supported by the Office of the Assistant Secretary of Defense for Health Affairs, through the Peer Reviewed Cancer Research Program, under Award No. W81XWH-17-1-0245. Opinions, interpretations, conclusions, and recommendations are those of the author and are not necessarily endorsed by the Department of Defense. PW was supported by the Damon Runyon Cancer Research Foundation (DRSG-28P-19). The Genetics Bioinformatics Service Center of Stanford Cancer Institute's Share Resource Facility was supported by NIH grant P30 CA124435.

## Additional information

### Competing interests

Roel Nusse: Reviewing editor, *eLife*. The other authors declare that no competing interests exist.

### Funding

| Funder | Grant reference number | Author |
|---|---|---|
| Howard Hughes Medical Institute | | Roel Nusse |
| Stinehart Reed Foundation | | Roel Nusse |
| Office of the Assistant Secretary of Defense for Health Affairs, through the Peer Reviewed Cancer Research Program | W81XWH-17-1-0245 | Abby Sarkar |
| Damon Runyon Cancer Research Foundation | DRSG-28P-19 | Peng Wu |
| National Institutes of Health | P30 CA124435 | Yan Yang |

The funders had no role in study design, data collection, and interpretation, or the decision to submit the work for publication.

### Author contributions

Abby Sarkar, Conceptualization, Resources, Formal analysis, Validation, Investigation, Visualization, Methodology, Writing - original draft, Writing – review and editing; Yinhua Jin, Huy Nguyen, Formal analysis, Investigation, Visualization, Methodology, Writing – review and editing; Brian C DeFelice, Resources, Data curation, Formal analysis, Investigation, Visualization, Methodology, Writing – review and editing; Catriona Y Logan, Formal analysis, Validation, Investigation, Visualization, Methodology, Writing – review and editing; Yan Yang, Resources, Formal analysis, Investigation, Visualization, Methodology, Writing – review and editing; Teni Anbarchian, Azalia M Martínez Jaimes, Formal analysis, Investigation, Methodology, Writing – review and editing; Peng Wu, Resources, Investigation, Methodology, Writing – review and editing; Maurizio Morri, Norma F Neff, Resources, Investigation, Methodology; Eric Rulifson, Resources, Methodology; Matthew Fish, Resources, Formal analysis, Investigation, Methodology; Avi Gurion Kaye, Formal analysis, Investigation, Methodology; Roel Nusse, Conceptualization, Resources, Supervision, Funding acquisition, Methodology, Writing - original draft, Writing – review and editing

### Author ORCIDs

Abby Sarkar  http://orcid.org/0000-0001-6101-1721
Peng Wu  http://orcid.org/0000-0001-6565-0002

Roel Nusse http://orcid.org/0000-0001-7082-3748

## Ethics

All animal experiments and methods were approved by the Institutional Animal Care and Use Committee at Stanford University.

## Decision letter and Author response

Decision letter https://doi.org/10.7554/eLife.82311.sa1
Author response https://doi.org/10.7554/eLife.82311.sa2

## Additional files

### Supplementary files

- Supplementary file 1. Metabolomics intermittent fasting (IF) versus ad libitum (AL significantly changed metabolites between IF and AL samples from metabolomic studies).
- MDAR checklist

### Data availability

Single-cell RNA sequencing has been deposited in GEO under accession number GSE211693. Metabolomics has been deposited in Metabolomics Workbench by NIH Common Fund's National Metabolomics Data Repository (NMDR); Project DOI: https://doi.org/10.21228/M8Z119.

The following datasets were generated:

| Author(s) | Year | Dataset title | Dataset URL | Database and Identifier |
|---|---|---|---|---|
| Sarkar A, Jin Y, DeFelice BC, Logan CY, Yan Y, Anbarchian T, Wu P, Morri M, Neff N, Nguyen H, Rulifson E, Fish M, Kaye AG, Martínez Jaimes AM, Nusse R | 2022 | Intermittent fasting induces rapid hepatocyte proliferation to restore the hepatostat in the mouse liver | https://www.ncbi.nlm.nih.gov/geo/query/acc.cgi?acc=GSE211693 | NCBI Gene Expression Omnibus, GSE211693 |
| Sarkar A, Jin Y, DeFelice BC, Logan CY, Yan Y, Anbarchian T, Wu P, Morri M, Neff N, Nguyen H, Rulifson E, Fish M, Kaye AG, Martínez Jaimes AM, Nusse R | 2022 | Intermittent fasting induces rapid hepatocyte proliferation to restore the hepatostat in the mouse liver | https://www.metabolomicsworkbench.org/data/DRCCMetadata.php?Mode=Project&ProjectID=PR001445 | Metabolomics Workbench, PR001445 |

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

# Appendix 1

**Appendix 1—key resources table**

| Reagent type (species) or resource | Designation | Source or reference | Identifiers | Additional information |
|---|---|---|---|---|
| gene (*Mus musculus*) | *Fgf15* | GenBank | *BC021328 cloneID 5066286* | Male |
| Strain, strain background (*Mus musculus, male*) | *C57BL/6 J* | The Jackson Laboratory | Cat# 000664 RRID:IMSR_JAX:000664 | |
| Strain, strain background (*Mus musculus, male*) | *Rosa26-CreERT2* | The Jackson Laboratory | Cat# 008463 RRID:IMSR_JAX:00846 | |
| Strain, strain background (*Mus musculus, male*) | *Rosa26-Confetti* | The Jackson Laboratory | Cat# 017492 RRID:IMSR_JAX:008463 | |
| Strain, strain background (*Mus musculus, male*) | Axin2-rtTA | The Jackson Laboratory | Cat# 016997 RRID:IMSR_JAX:016997 | |
| Strain, strain background (*Mus musculus, male*) | TetO-H2B-GFP | The Jackson Laboratory | Cat# 005104 RRID:IMSR_JAX: 005104 | |
| Strain, strain background (*Mus musculus, male*) | TetO-Cre | The Jackson Laboratory | Cat#006234 RRID:IMSR_JAX:006234 | |
| Strain, strain background (*Mus musculus, male*) | *Rosa26-mTmG* | The Jackson Laboratory | Cat# 037456 RRID:IMSR_JAX: 037456 | |
| Strain, strain background (*Mus musculus, male*) | *Rosa26-Cas9* | The Jackson Laboratory | Cat# 026179 RRID:IMSR_JAX: 026179 | |
| Strain, strain background (*Mus musculus, male*) | *Tbx3 flox* | Dr. Anne Moon | N/A | |
| Strain, strain background (*Mus musculus, male*) | *Klb* flox | Dr. David Mangelsdorf | N/A | |
| antibody | anti-GFP (Chicken polyclonal) | Abcam | Cat# ab13970 RRID:AB_300798 | 1:500 IF |
| antibody | Anti-RFP (Rabbit polyclonal) | Rockland | Cat# 600-401-379 RRID:AB_2209751 | 1:500 IF |
| antibody | Anti-Hnf4 (Mouse monoclonal) | Abcam | Cat# ab41898 RRID:AB_732976 | 1:500 IF; 50 IHC |
| antibody | Anti-Klotho (Rat monoclonal) | DSHB | Klotho KL-115 RRID:AB_2618099 | 1:50 IF |
| antibody | Anti-FGF15 (Mouse monoclonal, IgG2a) | Santa Cruz | sc-514647 RRID NA | 1:50 IF |
| antibody | Anti-Phospho-Tryosine (mouse monoclonal) | Cell Signaling Technology | Cat# 9411 RRID:AB_331228 | 1:50 IF |
| antibody | Anti-Phospho-c-Jun (Ser73) (rabbit monoclonal) | Cell Signaling Technology | Cat# 3270, RRID:AB_2129575 | 1:50 IF |
| antibody | Anti-Tbx3 (Goat polyclonal) | Santa Cruz Biotechnology | Cat# sc-17871 RRID:AB_661666 | 1:50 IHC |

*Appendix 1 Continued on next page*

*Appendix 1 Continued*

| Reagent type (species) or resource | Designation | Source or reference | Identifiers | Additional information |
|---|---|---|---|---|
| antibody | Anti-Goat IgG (Donkey polyclonal) | Jackson Immuno Research Labs | Cat# 705-065-147 RRID:AB_2340397 | 1:200 IHC |
| antibody | Anti-Glutamine Synthetase (Mouse monoclonal) | Millipore | Cat# MAB302 RRID:AB_2110656 | 1:500 IHC |
| antibody | Anti-Catenin, beta (mouse monoclonal) | BD Biosciences | Cat# 610154 RRID:AB_397555 | 1:50 IHC |
| antibody | Anti-Hnf4 (Rabbit polyclonal) | Santa Cruz Biotechnology | Cat# sc-8987 RRID:AB_2116913 | 1:50 IHC |
| antibody | KI67(SolA15) (Rat, monoclonal) | Thermo Fisher Scientific | Cat# 14-5698-82 RRID:AB_10854564 | 1:50 IHC |
| recombinant DNA reagent | pAAV-Guide-it-Down | *Clontech Laboratories Inc.* | Cat# *041315* | |
| recombinant DNA reagent | pscAAV-TTR-mFgf15 | This paper and Addgene | Currently Deposit 81516 | |
| sequence-based reagent | sgAPC_F | This paper | Assembly primers for pAAV-Guide-it-Down targeting | CCGGAGGCTGCATGAGAGCACTTG3 |
| sequence-based reagent | sgAPC_F | This paper | Assembly primers for pAAV-Guide-it-Down targeting | AAACCAAGTGCTCTCATGCAGCCT3 |
| sequence-based reagent | sgRNA: targeting Apc | This paper | Targeting sequence | AGGCTGCATGAGAGCACTTG |
| commercial assay or kit | In-Fusion HD Cloning | Clontech | Cat# 639647 | |
| commercial assay or kit | RNAscope probe-Mm-Cyp2f2 | Advanced Cell Diagnostics | Cat# 451851 | target region: 555–169 |
| commercial assay or kit | RNAscope probe-Mm-Cyp2e1-C2 | Advanced Cell Diagnostics | Cat# 402781 C2 | target region: 458–1530 |
| commercial assay or kit | RNeasy Mini Isolation Kit | Qiagen | Cat# 74004 | |
| commercial assay or kit | High Capacity cDNA Reverse Transcription Kit | Life Technologies | Cat# 4368814 | |
| commercial assay or kit | Taqman Gene Expression Assay (*Gapdh*) | ThermoFisher Scientific | Cat# 4331182; Mm99999915_g1 | |
| commercial assay or kit | Expression Assay (*Klb*) | ThermoFisher Scientific | Cat# 4331182; Mm00473122_m1 | |
| commercial assay or kit | Expression Assay (*Fgf15*) | ThermoFisher Scientific | Cat# 4331182; Mm00433278_m1 | |
| commercial assay or kit | Expression Assay (*Tbx3*) | ThermoFisher Scientific | Cat# 4331182; Mm01195719_m1 | |
| commercial assay or kit | Chromium Single Cell 3" Reagents Kit V3 | 10 x Genomics | | Discontinued |
| commercial assay or kit | NovaSeq S2 v.1.5 Reagent Kits | Illumnina | NA | Discontinued |

*Appendix 1 Continued on next page*

*Appendix 1 Continued*

| Reagent type (species) or resource | Designation | Source or reference | Identifiers | Additional information |
|---|---|---|---|---|
| commercial assay or kit | Filter Microplates | Agilent Technologies | Cat#203980–100 | |
| chemical compound, drug | Tamoxifen | Sigma Aldrich | Cat# T5648-1G | |
| chemical compound, drug | Doxycycline hyclate | Sigma-Aldrich | Cat# D9891 | |
| chemical compound, drug | HistoClear | Natural Diagnostics | Cat# HS2001GLL | |
| chemical compound, drug | Antigen Unmasking Solution, Tris-Based | Vector Labs | Cat# H-3301 | |
| chemical compound, drug | Avidin/Biotin Blocking Kit | Vector Labs | Cat# SP-2001 | |
| chemical compound, drug | Click-iT Plus TUNEL Assay Kits for In Situ Apoptosis Detection | ThermoFisher Scientific | Cat# C10619 | |
| chemical compound, drug | FxCycle PI/RNase | ThermoFisher Scientific | Cat# F10797 | |
| chemical compound, drug | TRIzol Reagent | Invitrogen | Cat# 15596026 | |
| software, algorithm | ImageJ | NIH https://imagej.net/ | RRID:SCR_003070 | |
| software, algorithm | GraphPad Prism 5.0 software | GraphPad Software; http://www.graphpad.com | RRID:SCR_002798 | |
| software, algorithm | *Cell Ranger Software (v3.1.0, mm10 ref genome)* | 10 x Genomics Software; https://support.10xgenomics.com/single-cell-gene-expression/software/pipelines/latest/what-is-cell-ranger | RRID:SCR_017344 | |
| software, algorithm | *Seraut Software (v3.0, R package)* | Seurat Software; https://satijalab.org/seurat/get_started.htm | RRID:SCR_016341 | |
| software, algorithm | BD FACS Diva 8.0 software (BD) | BD FACS Diva software; http://www.bdbiosciences.com/instruments/software/facsdiva/index.jsp | RRID:SCR_001456 | |
| software, algorithm | MS-DIAL v4.60 software | MS-DIAL software; (*Tsugawa et al., 2020*) | NA | |
| software, algorithm | MetaboAnalyst 5.0 software | MetaboAnalyst software; https://www.metaboanalyst.ca/ | RRID:SCR_015539 | |
| Other | AAV/DJ8-Ttr-Cre | *Vector Bio Labs* | 7102 | AAV-DJ8 virus that expresses an improved Cre under a liver-specific Ttr promoter |
| Other | AAV8-Null | *Vector Bio Labs* | 7077 | AAV serotype 8 virus that has a CMV promoter with no transgene. It's used as control AAV in the paper. |

