## [Editor Report]

This work reports that intermittent fasting alters the homeostatic regenerative programme with fundamental implications for the use of murine models to study liver regeneration and cancer and highlights through a series of solid mechanistic studies the role of FGF/Wnt signalling interactions in modulating fasted-associated regeneration. It opens up further questions as to why this occurs, how this is beneficial to adapting to a fasting state and how we should design and interpret preclinical animal studies.

---

## [Decision Letter]

**Decision letter after peer review:**

Thank you for submitting your article "Intermittent fasting induces rapid hepatocyte proliferation to restore the hepatostat in the mouse liver" for consideration by *eLife*. Your article has been reviewed by 2 peer reviewers, including Pramod Mistry as Reviewing Editor and Reviewer #1, and the evaluation has been overseen by Mone Zaidi as the Senior Editor. The following individual involved in the review of your submission has agreed to reveal their identity: Tom Bird (Reviewer #2).

Essential revisions:

This is an interesting and highly provocative report opening up a variety of questions relating to how we interpret mouse models in the context of their environment, here relating to feeding, in comparison to human physiology and pathophysiology. Overall, the authors are to be congratulated on this important body of work comprising a series of well-designed, rigorously performed, and appropriately analyzed experiments. The conclusions on the whole are strongly supported by the data presented. The fundamental question is topical and the results provocative.

Recommendations:

1. Regarding the conclusions on the timescale of proliferation relative to refeeding. The analysis in Figure 1 is taken 30 minutes following refeeding and this hyperproliferation is then related to FGF15 transcription in the intestine. Can the authors comment on how they reconcile the time frame of induction given their data? Is there downregulation of the downstream pathway over the same time course of FGF15 transcription in the intestine for example? Recommend nuance in the conclusions in the abstract to take account of the lack of direct evidence for intestinally derived FGF15.

2. The FGF/Wnt-Bcatenin interaction is compelling and shown robustly. Additional characterization of the APC model would be helpful, specifically a statement of whether those clones only outwith the physiological GS areas were quantified, whether these clones also express Tbx3 (and/or other Bcatenin targets) as would be anticipated and show relative hyperproliferation (measured by Ki67/BrdU) compared to other hepatocytes. It would also be helpful to test whether the IF status affects the baseline Wnt/Bcat signature across the liver lobule, either dependent or independent of FGF signaling via Klb. The pseudobulk transcriptomic data presented in Supplementary Figure 2 goes some way to reassuring that there is no such effect on zonal Wnt signatures but this would be further supported by quantified IHC data examining specific B-cat targets e.g. GS/OAT etc.

3. It is notable that the proliferation at 1-3 weeks IF is towards the inner border of GS-expressing hepatocytes. Do the authors know the relative expression of Tbx3 in these hepatocytes and do they have data to suggest that altered, specifically higher, levels of Tbx3 in the immediately pericentral hepatocytes may relatively impair proliferation – was this dependent on FGF levels for example in the AAV overexpression system?

4. Suggest caution in the use of Mann-Whitney tests to compare multiple comparisons of integer data (clonal size). A statistical comparison of the data in Supplementary Figure 2 would be welcomed.

5. Please note that the 3-week control IF data in Figures 4A and B are conflicting. Are the authors confident that the biological sample size is sufficient to demonstrate statistically valid interpretation from their depletion studies?

*Reviewer #2 (Recommendations for the authors):*

This is an interesting and highly provocative report opening up a variety of questions relating to how we interpret mouse models in the context of their environment, here relating to feeding, in comparison to human physiology and pathophysiology. Overall, the authors are to be congratulated on this important body of work compromising a series of well-designed, rigorously performed, and appropriately analysed experiments. The conclusions on the whole are strongly supported by the data presented. The fundamental question is topical and the results provocative.

My single largest concern regarding the conclusions drawn from the data presented is in relation to the timescale of proliferation relative to refeeding. The analysis in Figure 1 is taken 30 minutes following refeeding and this hyperproliferation is then related to FGF15 transcription in the intestine. This does not definitely prove that intestinally derived FGF15 is driving this proliferation. Secondly, regarding this temporal relationship, it seems remarkable to me that the appearance of transcript in the intestine would, within a matter of minutes, result in the endocrine induction of proliferation (Ki67 expression) in another organ. Can the authors comment on how they reconcile the time frame of induction given their data? Is there downregulation of the downstream pathway over the same time course of FGF15 transcription in the intestine for example? I do not feel that an intestinal knockout is required, but would recommend nuancing the conclusions in the abstract to take account of the lack of direct evidence for intestinally derived FGF15.

The FGF/Wnt-Bcatenin interaction and compelling and shown robustly. Additional characterisation of the APC model would be helpful, specifically a statement of whether those clones only outwith the physiological GS areas were quantified, whether these clones also express Tbx3 (and/or other Bcatenin targets) as would be anticipated and show relative hyperproliferation (measured by Ki67/BrdU) compared to other hepatocytes. It would also be helpful to test whether the IF status affects the baseline Wnt/Bcat signature across the liver lobule, either dependent or independent of FGF signaling via Klb. The pseudobulk transcriptomic data presented in Supplementary Figure 2 goes some way to reassuring that there is no such effect on zonal Wnt signatures but this would be further supported by quantified IHC data examining specific B-cat targets e.g. GS/OAT etc.

It is notable that the proliferation at 1-3 weeks IF is towards the inner border of GS-expressing hepatocytes. Do the authors know the relative expression of Tbx3 in these hepatocytes and do they have data to suggest that altered, specifically higher, levels of Tbx3 in the immediately pericentral hepatocytes may relatively impair proliferation – was this dependent on FGF levels for example in the AAV overexpression system?

I would caution against the use of Mann-Whitney tests to compare multiple comparisons of integer data (clonal size).

Statistical comparison of the data in Supplementary Figure 2 would be welcomed.

I would note that the 3-week control IF data in Figures 4A and B are conflicting. Are the authors confident that the biological sample size is sufficient to demonstrate statistically valid interpretation from their depletion studies?

---

## [Author Response]

Recommendations:1. Regarding the conclusions on the timescale of proliferation relative to refeeding. The analysis in Figure 1 is taken 30 minutes following refeeding and this hyperproliferation is then related to FGF15 transcription in the intestine. Can the authors comment on how they reconcile the time frame of induction given their data? Is there downregulation of the downstream pathway over the same time course of FGF15 transcription in the intestine for example? Recommend nuance in the conclusions in the abstract to take account of the lack of direct evidence for intestinally derived FGF15.

We thank the reviewers for this feedback. We observed a significant increase in hepatocyte proliferation, by Ki67+ expression, 30 mins after re-feeding in 1-week IF treated animals, compared to AL treated animals and other IF timepoints. During these same time frames, we observed a corresponding peak in Fgf15 expression in the intestine and activation of FGF15-bKLOTHO signaling in the liver at 30 mins post re-feeding.

We agree with reviewers that although our experiments provide direct, functional evidence that liver FGF15-bKLOTHO signaling regulates hepatocyte proliferation, our results connecting intestinal-produced FGF15 and hepatocyte proliferation are descriptive. We therefore have taken the reviewers recommendations and adjusted the conclusions in the manuscript abstract and results. We hope that future studies will unveil if there is inter-organ, intestine-liver regulation of hepatocyte proliferation during IF.

2. The FGF/Wnt-Bcatenin interaction is compelling and shown robustly. Additional characterization of the APC model would be helpful, specifically a statement of whether those clones only outwith the physiological GS areas were quantified, whether these clones also express Tbx3 (and/or other Bcatenin targets) as would be anticipated and show relative hyperproliferation (measured by Ki67/BrdU) compared to other hepatocytes.

We thank the reviewers for asking for this important clarification. Indeed, only GS+ Apc mutant clones that had discrete boundaries from physiological GS+ pericentral cells were quantified in Figure 3A. Boundaries were determined by having one or more GS negative hepatocytes between the GS+ Apc mutant clone and physiological GS+ pericentral cells. We have clarified this in the methods section of the manuscript.

Evidence demonstrating that non-pericentral GS+ cells are Tbx3+ in the AAV-U6-sgAPC model has been described in a previous manuscript from our group (Jin et al. 2022). Evidence demonstrating Apc mutated GS+ hepatocytes are hyperproliferative compared to other hepatocytes has been described in (Benhamouche et al., 2006). Our work here has focused on comparing the clonal expansion of Apc mutant cells in the context of IF vs AL. We find that clonal expansion is enhanced with IF (Figure 3A).

It would also be helpful to test whether the IF status affects the baseline Wnt/Bcat signature across the liver lobule, either dependent or independent of FGF signaling via Klb. The pseudobulk transcriptomic data presented in Supplementary Figure 2 goes some way to reassuring that there is no such effect on zonal Wnt signatures but this would be further supported by quantified IHC data examining specific B-cat targets e.g. GS/OAT etc.

To address the reviewer’s request, we have included Author response image 1 expression analyses of Axin2, a robust downstream target of Wnt signaling, to demonstrate the impact that 1 week IF treatment has on baseline Wnt/Bcat signaling. We did not observe a baseline change in Axin2 expression between feeding regimens, AL vs. IF, nor a change in lobule distribution of Axin2 (data not shown). We also did not observe a change in Axin2 expression in the presence or absence of liver Klb/Fgf15 signaling.

**Author response image 1. sa2fig1:** 

3. It is notable that the proliferation at 1-3 weeks IF is towards the inner border of GS-expressing hepatocytes. Do the authors know the relative expression of Tbx3 in these hepatocytes and do they have data to suggest that altered, specifically higher, levels of Tbx3 in the immediately pericentral hepatocytes may relatively impair proliferation – was this dependent on FGF levels for example in the AAV overexpression system?

We thank the reviewers for this comment. The reviewers’ hypothesis that hepatocytes with higher levels of Tbx3 may have impaired proliferation, rather than enhanced proliferation, is an intriguing one. Understanding the impact of Tbx3 levels/dosage on hepatocyte proliferative is of great interest. Previous studies using in vitro experiments have demonstrated that higher levels of Tbx3 lead to greater repression of cell cycle inhibitors and consequential enhancement of cell proliferation (Khan et al., 2020; Jin et al., 2022). However, previous studies in the liver have indicated that GS+ hepatocytes, hepatocytes known to have the highest Tbx3 expression across the liver lobule, are less proliferative during AL feeding conditions than other hepatocytes (Wei et al., 2021; He et al., 2021).This suggests that perhaps there are other regulators, besides Tbx3, in GS+ cells that are anti-proliferative and that these regulators may also be impacted by IF treatment. What these regulators are and how they repress hepatocyte proliferation remains an important question for future studies.

4. Suggest caution in the use of Mann-Whitney tests to compare multiple comparisons of integer data (clonal size). A statistical comparison of the data in Supplementary Figure 2 would be welcomed.

As the reviewers suggested, we have replaced the Mann-Whitney test with 1-way ANOVA in Supplementary Figure 2.

5. Please note that the 3-week control IF data in Figures 4A and B are conflicting. Are the authors confident that the biological sample size is sufficient to demonstrate statistically valid interpretation from their depletion studies?

We thank the reviewer for observing this. We calculated % Liver to Body Weight for Figure 4A and Figure 4B using an N=7 and N=5, respectively. We believe the differences in precents among these two distinct experiments may have to do with mouse background. Figure 4A was performed on C57/Blk6 male mice. Figure 4B was performed on littermates from Klb KO; Tbx3 KO animals.